# Antibody Watch: Text mining antibody specificity from the literature

**Chun-Nan Hsu**[1], **Chia-Hui Chang**[1,2], **Thamolwan Poopradubsil**[2], **Amanda Lo**[1], **Karen A. William**[1], **Ko-Wei Lin**[1], **Anita Bandrowski**[1,3], **Ibrahim Burak Ozyurt**[1], **Jeffrey S. Grethe**[1], **Maryann E. Martone**[1,3] *

**1** Department of Neurosciences and Center for Research in Biological Systems, University of California, San Diego, La Jolla, California, United States of America, **2** Department of Computer Science and Information Engineering, National Central University, Zhongli, Taiwan, **3** SciCrunch, Inc. San Diego, California, United States of America

* mmartone@ucsd.edu

**Data Availability Statement:** All dataset files are available at Zenodo: doi:10.5281/zenodo.3701930.

**Funding:** C.-H. C. and T.P. were supported by the Ministry of Science and Technology, Taiwan under grant 108-2918-I-008-004. Other authors (C.-N. H,

## Abstract

Antibodies are widely used reagents to test for expression of proteins and other antigens. However, they might not always reliably produce results when they do not specifically bind to the target proteins that their providers designed them for, leading to unreliable research results. While many proposals have been developed to deal with the problem of antibody specificity, it is still challenging to cover the millions of antibodies that are available to researchers. In this study, we investigate the feasibility of automatically generating alerts to users of problematic antibodies by extracting statements about antibody specificity reported in the literature. The extracted alerts can be used to construct an "Antibody Watch" knowledge base containing supporting statements of problematic antibodies. We developed a deep neural network system and tested its performance with a corpus of more than two thousand articles that reported uses of antibodies. We divided the problem into two tasks. Given an input article, the first task is to identify snippets about antibody specificity and classify if the snippets report that any antibody exhibits non-specificity, and thus is problematic. The second task is to link each of these snippets to one or more antibodies mentioned in the snippet. The experimental evaluation shows that our system can accurately perform the classification task with 0.925 weighted F1-score, linking with 0.962 accuracy, and 0.914 weighted F1 when combined to complete the joint task. We leveraged Research Resource Identifiers (RRID) to precisely identify antibodies linked to the extracted specificity snippets. The result shows that it is feasible to construct a reliable knowledge base about problematic antibodies by text mining.

## Author summary

Antibodies are widely used reagents to test for the expression of proteins. However, antibodies are also a known source of reproducibility problems in biomedicine, as specificity and other issues can complicate their use. Information about how antibodies perform for specific applications are scattered across the biomedical literature and multiple websites. To alert scientists with reported antibody issues, we develop text mining algorithms that can identify

A. L., K. W., K.-W. L., A. B., I. B. O., J. S. G., and M. E. M.) have been supported by NIH grant U24DK097771, which supports the National Institute of Diabetes and Digestive and Kidney Diseases (NIDDK) Information Network (dkNET, https://dknet.org), and NIH's National Institute on Drug Abuse award U24DA039832, which supports the Neuroscience Information Framework (http://neuinfo.org). The funders had no role in study design, data collection and analysis, decision to publish, or preparation of the manuscript.

**Competing interests:** I have read the journal's policy and the authors of this manuscript have the following competing interests: M.E.M, J.S.G., A.B. have an equity interest in SciCrunch, Inc., a company that may potentially benefit from the research results. The terms of this arrangement have been reviewed and approved by the University of California, San Diego in accordance with its conflict of interest policies.

specificity issues reported in the literature. We developed a deep neural network algorithm and performed a feasibility study on 2,223 papers. We leveraged Research Resource Identifiers (RRIDs), unique identifiers for antibodies and other biomedical resources, to match extracted specificity issues with particular antibodies. The results show that our system, called "Antibody Watch," can accurately perform specificity issue identification and RRID association with a weighted F-score over 0.914. From our test corpus, we identified 37 antibodies with 68 nonspecific issue statements. With Antibody Watch, for example, if one were looking for an antibody targeting beta-Amyloid 1–16, from 74 antibodies at dkNET Resource Reports (on 10/2/20), one would be alerted that "some non-specific bands were detected at 55 kDa in both WT and APP/PS1 mice with the 6E10 antibody. . ."

This is a *PLOS Computational Biology* Software paper.

## Introduction

Antibodies are some of the most common and powerful experimental reagents for detection and localization of proteins, peptides, polysaccharides, and other antigens. They are essential in biomedical research. In an immune system, an antibody binds to an antigen with the top tips of its Y-shaped chemical structure. Scientists take advantage of this property to design antibodies that target specific antigens to detect their presence or to isolate them from a mixture. Examples of common antibody assays include immunohistochemistry (IHC [1]), western blot (WB [2]), flow cytometry (FC [3]), and enzyme-linked immunosorbent assay (ELISA [4]). Antibodies are playing an important role in studies of COVID-19. Over 281 unique antibodies are associated with COVID-19, according to the Antibody Registry (https://urldefense.com/v3/__https://antibodyregistry.org/covid19__;!!Mih3wA!RnOtVKTVZZE7zGET2GaOzjvbDlpQCch6_MCSP2lkt0gMeBkRB_qS2wwkvT8CbeB5I8g$).

One of the most common antibody problems is that the antibody may not bind only to the antigen that their providers design them for, known as the problem of *antibody specificity* [5, 6]. There may be cross reactivity due to design flaws, contamination, or use in a novel context where a similar antigen is detected. While clinical antibodies can be tested for the exact application for which they are indicated, research-use only antibodies will be tested typically in one or more applications in one or a small handful of species, but it is nearly impossible to test them in all species or under all conditions in which they can potentially be used. Therefore, these reagents, while specific in a mouse colon, for example, may have specificity issues when tested for the same target antigen in a zebrafish brain. An experiment using a nonspecific antibody may lead to misinterpretation of the results, leading to inaccurate conclusions.

Many have proposed systematic validation of antibodies, including experimental validation by large consortium projects such as the Antibody Validation Database [7] from the ENCODE project [8] or the independent validation project [9]; feedback collection, such as the BioCompare Antibody Search Tool (https://urldefense.com/v3/__https://www.biocompare.com/Antibodies__;!!Mih3wA!RnOtVKTVZZE7zGET2GaOzjvbDlpQCch6_MCSP2lkt0gMeBkRB_qS2wwkvT8C2ssCbHc$) and AntibodyPedia (https://urldefense.com/v3/__https://www.antibodypedia.com__;!!Mih3wA!RnOtVKTVZZE7zGET2GaOzjvbDlpQCch6_MCSP2lkt0gMeBkRB_qS2wwkvT8CYTdv94o$); and curation based on figures of antibody

experimental results reported in the literature, such as BenchSci (https://urldefense.com/v3/__ https://www.benchsci.com__;!!Mih3wA!RnOtVKTVZZE7zGET2GaOzjvbDlpQCch6_ MCSP2lkt0gMeBkRB_qS2wwkvT8CpEsKQ88$). However, because there are more than two million unique antibodies, according to the Antibody Registry, and new ones are constantly created, it is still challenging to cover all antibodies with known specificity issues.

Many problems with antibodies are only encountered in the context of individual studies, where authors work to validate antibodies before use and may report problems with some reagents. We propose to address the antibody specificity problem by constructing a knowledge base containing statements about antibody specificity automatically extracted from the biomedical literature. A large number of statements about antibody specificity are available by authors who used antibodies in their experiments. Automated text mining techniques can be applied to a large corpus of publications to extract and disseminate the information automatically at a pace that matches the growth of new antibodies and publications and provide scientists up-to-date alerts of problematic antibodies to assist their selection of antibodies. The key contributions of this work include:

- We propose a novel approach to the problem of antibody specificity by alerting scientists if an antibody is well validated or may be problematic as reported in the literature. The problem is important in helping ensure reliability of studies using antibodies in the experiments.

- We show that the approach is feasible by developing an automated text mining system called $(ABSA)^2$ and empirically evaluate its performance with an in-house annotated corpus of $\sim 2,000$ articles. $(ABSA)^2$, which stands for <u>A</u>nti<u>B</u>ody <u>S</u>pecificity <u>A</u>nnotator by <u>A</u>spect-<u>B</u>ased <u>S</u>entiment <u>A</u>nalysis, is a deep neural network model that distinguishes specificity of antibodies stated in a snippet. $(ABSA)^2$ achieves the best F-score for the task of identifying problematic antibodies in our experimental evaluation, outperforming all baselines and competing models.

- We show that with our automated text mining system, combining author-supplied Research Resource Identifier (RRID) [10–13] with advanced deep neural network Natural Language Processing (NLP), we can unambiguously identify an antibody mentioned in the literature, allowing us to link an antibody specificity statement automatically extracted from the literature with an exact antibody referred to by the statement. This is crucial in order to provide useful alerts of problematic antibodies. We anticipate that similar RRID-NLP hybrid text mining approaches can be applied to quantify qualities and appropriate usages of biomedical research resources covered by the RRID, including, *e.g.*, cell lines, bioinformatics tools, data repositories, *etc.*, and properly credit developers of these research resources, a long standing issue of modern biomedical research that depends heavily on research resources [13–17].

## Materials and methods

Our goal is to construct a knowledge base called "Antibody Watch" as a part of our "Resource Watch" services in dkNET (https://urldefense.com/v3/__https://dknet.org__;!!Mih3wA! RnOtVKTVZZE7zGET2GaOzjvbDlpQCch6_MCSP2lkt0gMeBkRB_qS2wwkvT8CPUGvyhY $) for scientist users to check if the antibody they are interested in has been reported in the literature and if any information has been provided about its specificity with regard to its designated target antigen. In our vision, "Resource Watch" will cover a broad range of biomedical research resources in addition to antibodies, such as cell lines, model organisms, bioinformatics tools, and data repositories, *etc.*"Antibody Watch" will focus on antibodies and provide, for

**Table 1. Example snippets of the antibody specificity classes and the PubMed IDs (PMID) of their sources.**

| Class | Example | PMID |
|---|---|---|
| Nonspecific (Negative) | Some *non−specific* bands were detected at ~55 kDa in both WT and APP/PS1 mice with the 6E10 *antibody* . . . | 30177812 |
| Specific (Positive) | Our *antibody* is *specific*, as each immunizing peptide blocked the corresponding immunoreactivity . . . | 25650666 |
| Neutral | Probing protein arrays with *antibodies* allows the assessment of their *specificity and cross−reactivity* across a large numbers of potential antigens in parallel . . . | 27335636 |

each unique antibody, a list of statements extracted from the literature about its specificity, along with metadata about the antibody to facilitate search.

## Problem formulation

Table 1 shows example snippets from real publications that contain keywords related to "antibody" (colored in red) and "specific" (in blue) and are potentially the statements that we would like to extract to include in our knowledge base.

In this work, we consider snippets consisting of no more than three sentences. The choice of three sentences at most is based on our assumption that the immediate adjacent sentences before and after the main sentences contain the keywords are sufficient to provide useful information while not bringing in too much irrelevant information that creates unwanted noise to burden the model. The assumption allows us to concentrate on training the model to make a classification based on three sentences in the context at a time instead of having to read the entire article.

We classify such snippets into one of the following classes: *nonspecific*, *specific* and *neutral*. Those snippets classified as nonspecific and specific will be included in the knowledge base while `neutral` ones will be excluded. Their definitions are:

- Nonspecific (Negative): the snippet states that an antibody may not always be specific to its target antigen and thus problematic.

- Specific (Positive): the snippet states that an antibody is specific to its target antigen.

- Neutral: all other snippets that are not about whether the antibody is specific to its target antigen and thus irrelevant for our purposes.

The problem is related to *sentiment analysis* that has been intensively studied in NLP, driven by the need of automatically classifying the sentiment of online customer product/service reviews. State-of-the-art approaches can classify not only the overall sentiment of a review but the sentiment of a designated aspect (*e.g.*, "appetizer" of restaurants) by leveraging attention mechanisms of deep neural networks (*e.g.*, [18–20]). We leveraged these ideas to develop effective classifiers for our antibody specificity classification task.

In the last decade, many approaches to aspect-based sentiment analysis (ABSA) have been developed ranging from statistical machine learning methods [21] to deep learning models [22]. For example, Wang *et al.* [23] developed an attention-based model with the bi-directional long-short term memory (BiLSTM) architecture [24]. Huang *et al.* [18] introduced an attention-over-attention (AOA) neural network to capture the interaction between aspects and context sentences. The AOA model outperformed previous BiLSTM-based architectures. A comprehensive survey of ABSA can be found in [22].

Once a snippet is extracted and classified as an antibody specificity statement, we need to link the statement to the exact antibody entities referred to in the statement. The task is

challenging because antibody references are obscure [5, 25]. Many antibody providers exist, including commercial suppliers and academic research labs of various sizes. For any given antigen, there can be many antibodies from different suppliers, derived from a wide range of organisms. For example, there are 136 antibodies in the Antibody Registry that match "6E10" mentioned in the first example snippet in Table 1, but that is the only clue about the antibody in that snippet.

Instead, we can search for detailed information about the antibodies in the same paper where the study design is described, usually appearing in the "Materials and Methods" section of the paper. For example, we have this sentence

*Purified anti-β-Amyloid, 1–16* antibody *(6E10) (Cat. No. 803003;* RRID:AB_2564652 *) was obtained from . . .*

(PMID 30177812)

in the paper that allows us to link the statement with "6E10" to this unique antibody entity. "PMID" here is the PubMed ID of the paper where the example snippet appears.

However, several issues must be resolved for the above idea of linking to work. First, how to identify snippets that contain detailed antibody information? Next, a study may involve several antibodies. How to use limited clues to correctly link to the right antibody? This is a special case of the coreference resolution problem in Natural Language Processing [26, 27], a challenging open research problem. Finally, the information may still be too obscure to allow correct identification of the exact antibody used. As reported by Vasilevsky *et al.* [14], in many cases, even the authors cannot recall exactly which antibody they have used, though the supplier name and the catalog ID were provided to identify an antibody. To identify the supplier, the city where the supplier is located may also be provided. The catalog IDs may become obsolete, authors may use a short-hand syntax for supplier catalog numbers, and suppliers may change names or merge with another company. Without sufficient metadata, neither NLP nor human experts will be able to infer the exact antibody from the text because the required information to correctly identify the antibody is simply not there.

All these issues can be resolved with the use of Research Resource Identifiers (RRID) [10]. Precisely, we will locate RRIDs of antibodies given in the paper and use the context of each RRID to guide linking of each antibody specificity statement to the exact one or more antibody entities that it refers to. In the example snippet above, "RRID:AB_2564652" is the RRID of the 6E10 antibody. Without it, uniquely identifying this antibody would still be difficult based only on other information such as the catalog number "Cat. No. 803003."

RRIDs were developed to identify and track what research resources were used in a given study. The antibody portion of the RRIDs is based on the Antibody Registry (https://urldefense.com/v3/__https://antibodyregistry.org/__;!!Mih3wA! RnOtVKTVZZE7zGET2GaOzjvbDlpQCch6_MCSP2lkt0gMeBkRB_qS2wwkvT8CqsOwlIw$), which assigns unique and persistent identifiers to each antibody so that they can be referenced within publications. Unlike antibody names or catalog numbers, these identifiers only point to a single antibody, so that the actual antibody used can be identified by humans, search engines, and automated agents. The identifier can be quickly traced back in the Antibody Registry. Once an antibody is assigned an identifier and entered into the Antibody Registry, the record will never be deleted, so even when an antibody disappears from a vendor's catalog or is sold to another vendor, the provenance of that antibody can still be traced. The two million antibody RRIDs currently in the Antibody Registry include commercial antibodies from about 300 vendors and non-commercial antibodies from more than 2,000 laboratories.

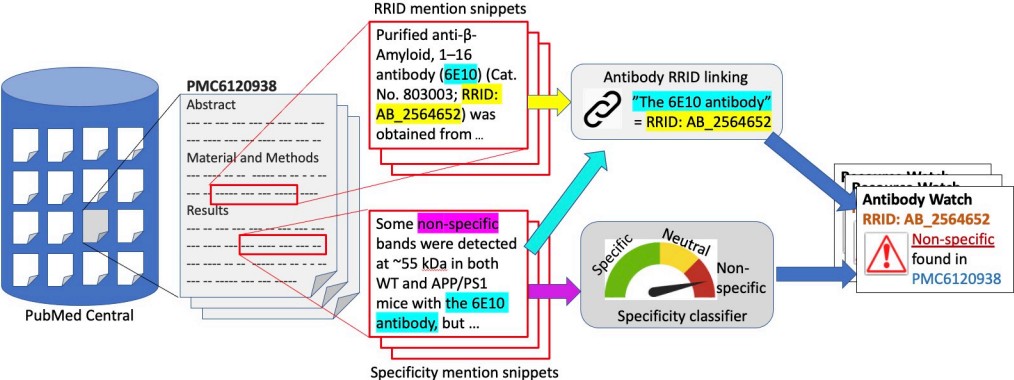

**Fig 1. The workflow to construct Antibody Watch.** Given the article PMC6120938, a set of "RRID mention snippets" and "Specificity mention snippets" will be extracted. Next, a "Specificity classifier" will determine if a specificity mention snippet states that the antibody, in this case "the 6E10 antibody," is specific to its target antigen or not. Then, "Antibody RRID linking" will link each specificity snippet to the "RRID mention snippets," and thus to one or more exact antibodies. In this example, "the 6E10 antibody" is linked to the antibody with "RRID:AB_2564652," which uniquely identifies an antibody. Finally, an entry is generated and entered into the Antibody Watch knowledge base to alert scientists that this antibody was reported to be nonspecific in PMC6120938 (PMID 30177812).

Authors supply RRIDs before their manuscript is published. Over 120 journals now request RRIDs to be included as part of their instructions to authors (*e.g.*, Cell Press, *eLife*, *the Journal of Neuroscience*, *Endocrinology*, to name a few). Examples of resources that can be identified using RRIDs include antibodies, cell lines, plasmids, model organisms and digital tools such as databases, bioinformatics software for statistics and data analysis, and any digital resources used in the research workflow [28].

If an antibody specificity statement cannot be linked to any RRID in the paper, then the statement is most likely about an antibody that is not used. This is one of the benefits of linking to RRIDs because authors are instructed by the publishers to specify an RRID only when the research resource was actually used in their study.

Fig 1 illustrates the complete workflow of our approach. Given a full-text article from a corpus, *e.g.*, PubMed Central, the workflow starts by extracting two types of snippets in the article. One of the types consists of "RRID mention snippets" that may appear in "Materials and Methods" section. The other type, "Specificity mention snippets," consists of statements about the specificity of the antibodies, usually appearing in the "Results" or "Discussion" sections, and figure/table legends. After these snippets were extracted, two tasks are applied to each type of snippet:

- Task 1 (Specificity classification) determines if a specificity snippet states that the antibody is specific or not with a deep neural network classifier.

- Task 2 (RRID linking) links each specificity statement to the exact antibody RRID(s) that it refers to.

The output knowledge base will contain, for each entry, a triple of an antibody, its specificity class (*i.e.*, specific or not), and the evidence—the snippet of the specificity statement in a source article.

## Algorithms

**Task 1: Specificity classification.** Task 1 is a three-class (positive, neutral, negative) categorization problem given a snippet related to antibody specificity. Our solution (ABSA)[2] is

inspired from models developed for *aspect-based sentiment analysis* (ABSA), a well-studied NLP problem aimed at identifying fine-grained opinion polarity for a given aspect. Similar to formulations of many ABSA models, in addition to the input snippet, (ABSA)$^2$ also takes the antibody term mentioned in the input snippet as our target aspect to capture specificity expressed towards each antibody. Specifying a target antibody is important also because there may be more than one antibody mentioned in a snippet where their specificity may be stated differently.

In particular, (ABSA)$^2$ is an attention-over-attention model (AOA) for ABSA [18], which was originally designed on top of context-independent word embeddings based on GLoVe [29]. AOA worked particularly well with BERT [30] (RRID:SCR_018008) and other contextualized word embedding transformers for our task over other competing ABSA models in our experimental investigations. Fig 2 shows the architecture of our model.

Given an input snippet $s = (w_1, w_2, \ldots, w_n)$ of $n$ tokens and the $m$ target aspect tokens $t = (x_1, \ldots, x_m)$ (*e.g.*, "antibody"). Our model first employs a BERT component with $L$ transformer layers to calculate the corresponding contextualized representations for the input token sequences with the form $H^0 = (\texttt{[CLS]}, s, \texttt{[SEP]}, t)$. Let $H^l$ be the output of the transformer at layer $l$, thus $H^l = (h_0^l, h_1^l, \ldots, h_n^l, h_{n+1}^l, \ldots, h_{n+m+1}^l)$ can be calculated by

$$H^l = \text{Transformer}(H^{l-1}).$$

Let $a = (h_1^L, \ldots, h_n^L)$ and $b = (h_{n+2}^L, \ldots, h_{n+m+1}^L)$ from the transformer's output at the last layer. The AOA model works by first calculating a pair-wise interaction matrix $I = a \cdot b^{\mathrm{T}} \in \mathbb{R}^{n \times m}$, where the value of each entry $I_{ij}$ represents the correlation of a word pair among the snippet and target. Let $\alpha$ be the column-wise softmax of $I$, representing the target-to-snippet attention, and $\beta$ be the row-wise softmax of $I$, representing the snippet-to-target attention:

$$\alpha_{ij} = \frac{exp(I_{ij})}{\sum_i^n exp(I_{ij})}, \beta_{ij} = \frac{exp(I_{ij})}{\sum_j^m exp(I_{ij})}.$$

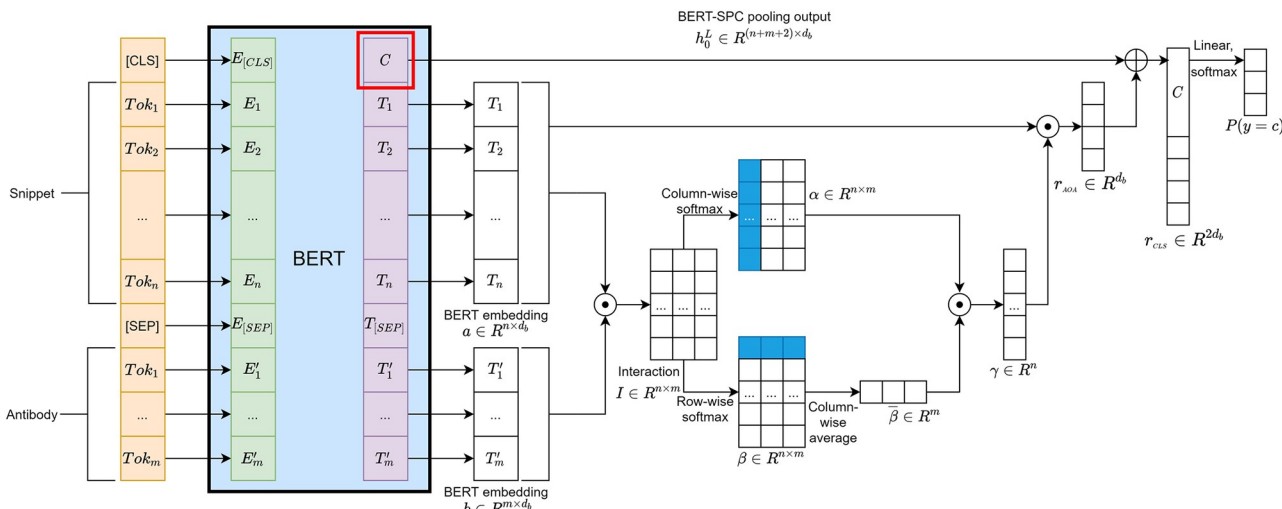

**Fig 2. A schematic diagram of the neural network architecture of (ABSA)$^2$ for classifying antibody specificity snippets.** (ABSA)$^2$ is an attention-over-attention model (AOA) based on ABSA [18] but with a transformer (left) as its input word embedding layer. (ABSA)$^2$ takes a snippet and an aspect token "antibody" as the input to classify the snippet into one of the specificity classes.

The idea of AOA is to compute the attention weight over averaged attention weight $\bar{\beta} \in \mathbb{R}^{1 \times m}$ by $\gamma = \alpha \bar{\beta}^{\mathrm{T}} \in \mathbb{R}^n$, where

$$\bar{\beta}_j \quad = \frac{1}{n} \sum_i^n \beta_{ij}.$$

Then AOA computes an attention-weighted representation of the input by

$$r_{\mathrm{AOA}} = a^{\mathrm{T}} \cdot \gamma. \tag{1}$$

Since the contextualized sentence-level representation from the last transformer layer $h_0^L$ usually provides useful information to a classification task, we concatenate $h_0^L$ with $r_{\mathrm{AOA}}$ as the final representation of the snippet:

$$r_{\mathrm{CLS}} = \mathrm{concatenate}(h_0^L, r_{\mathrm{AOA}}). \tag{2}$$

The final prediction layer is simply a linear layer that takes either $r_{\mathrm{CLS}}$ or $r_{\mathrm{AOA}}$ as the input and then followed by a softmax layer to obtain the class conditional probability for each class. The entire model is fine-tuned with a standard cross-entropy loss with $l^2$ regularization.

We tested many variants of the model described above including adding a layer of BiLSTM or multi-head attention (MHA) [20] on top of the transformer. As baselines, we also considered directly use a transformer with or without the aspect terms. The transformers that we compared include BERT and SciBERT [31]. Other transformers, such as BioBERT [32], can also be considered here.

Recently, models based on the attention-encoder network (AEN) [20] and local context focus mechanism (LCF) models [19] were reported to achieve state-of-the-art results for benchmark ABSA datasets [33]. We also implemented these models and their variants for a comparison. AEN learns to build attention links within an input sentence and between the input sentence and the aspects to perform aspect-based sentiment analysis. LCF focuses on using attention mechanisms to relate local context to the aspects. The details of our implementation are given in S1(A) Text.

**Task 2: Antibody RRID linking.** Task 2 is about linking an antibody specificity snippet to a candidate antibody RRID described in the same paper. We leveraged a BERT model as a sentence pair binary classifier (BERT-SPC) to determine if linking an input specificity snippet to a RRID snippet can be established. We used [SEP] token to split the specificity snippet and the RRID snippet and the output [CLS] as the prediction, that is, again, the standard formulation of a BERT-SPC model. We also considered SciBERT-SPC by replacing BERT with SciBERT. Again, other transformers, such as BioBERT, can be considered.

We implemented several baselines for the purpose of comparison. One of the baselines essentially counts common non-dictionary words in the two input snippets, *e.g.*, "6E10" in Fig 1. The common words were matched using the Jaro-Winkler distance [34]. We considered three thresholds: 1.0, 0.9, and 0.8. Another baseline is the Siamese Recurrent Architectures [35] with a BiLSTM layer. A neural network model with a Siamese Recurrent Architecture consists of two recurrent neural network components trained to have identical parameters (thus the name "Siamese") to read a pair of different input sentences. Then the difference between the output vectors of the two components is computed to determine if the input sentences are similar. We used either Manhattan or Euclidean distance as the last layer to measure the similarity between the input snippet pair. The details of our implementation are given in S1(B) Text.

### Implementation

**Dataset preparation.**  A corpus of 3,845 papers containing at least an antibody RRID in the regex pattern `AB_[0-9]`$^{+}$ identified by the RRID curation team on their publishers' websites were retrieved manually [28] and 47,403 RRIDs and their context snippets were extracted. Among these papers, 2,223 are in the PubMed Central Open Access subset and available legally for text mining. From the 2,223 full-text documents, we parsed each document into sections and split each section into sentences using the NLTK sentence splitter [36]. Then we selected the sentences which contain the regex patterns of `(S|s)pecific`, `((B|b)ackground staining)` or `(C|c)ross(|-)reactiv`, to extract the antibody specificity snippets. These patterns cover the terms used when antibody specificity is discussed and serve as a minimum filter to remove irrelevant snippets which may easily overwhelm the model. Other terms can be easily included to the data preparation steps if they are found to be useful.

To create snippets, the sentence containing these regex patterns is placed in the middle of the snippet surrounded by either previous or next sentence from the context. Each snippet contains at most three sentences, depending on where the key regex patterns appear. Those appearing in the figure legend of figures or the boundary of a paragraph may consist of less than 3 sentences. The number of snippets we obtained from this process was 22,013, from which we then chose 3,192 snippets (from 995 unique articles) that contained the regex pattern `(A|a)ntibod(y|ies)` for further labelling. The above steps basically extract candidate snippets that refer to antibody specificity. We examined the rest of 19,203 snippets and could not find any related to antibody specificity.

The snippets containing RRID mentions were extracted similarly but extra steps were performed to ensure that there will be no more than one RRID in an RRID snippet to be paired with a specificity snippet. Given a mention of RRID in the text, we include at most three sentences in its context to create the snippet in the same way as when we prepare snippets for specificity statements. However, if the sentence before or after the sentence where the RRID mention appears contain other RRID mentions, they will not be included. If the sentence where the RRID mention appears contains other RRID mentions, the sentence will be split into sub-strings at RRID mentions and only the sub-string that contains the RRID mention will be retained. The sub-strings that contain other RRID mentions will be discarded.

Since the format of RRID is strictly standardized, identifying RRID mentions is highly accurate. We have been using a semi-automatic curation pipeline to monitor how authors specify RRIDs in publications. In our previous study [28], we found that regex (RRID-by-RDW) achieved 0.944 F1-score. The errors are due to malformed RRIDs by authors. As more tools are developed to support the use of RRIDs, we expect these errors to diminish. An example of such tools has been implemented by the journal eLife [28].

**Data annotation.**  After initial labeling of the specificity levels and RRID linkings between the RRID and antibody snippets, we created a specialized annotation interface within a Google Sheets (RRID:SCR_017679) to further annotate the training examples. In each row, the spreadsheet interface listed the RRID snippet, followed by the supposed antibody snippet in correlation to the RRID, and Apps Script powered cells that allowed the curator to mark whether or not the RRID linking existed and the specificity of the snippets to one another. The RRID linking choices included yes and no, while the specificity labels included positive, neutral, and negative. As an example use case, if the antibody in the antibody snippet matched the RRID presented in the RRID snippet, then the RRID linking would be marked as "yes." And if the antibody snippet mentioned that the antibody was specific, then the specificity label would be marked as "positive."

**Table 2. Statistics of the annotated data.** We annotated 2,639 snippets for specificity classification (Task 1) and 7,245 snippet pairs for RRID linking (Task 2). Among 7,245, 1,100 are linked. Their specificity class distribution is shown (Joint).

| Task (unit) | Label Number | | | |
|---|---|---|---|---|
| Task 1: Specificity Classification (snippets) | Nonspecific | Neutral | Specific | Total |
| | 266 | 263 | 2,110 | 2,639 |
| Task 2: RRID Linking (snippet pairs) | Yes | No | | Total |
| | 1,100 | 6,145 | | 7,245 |
| Joint (triples of RRID-specificity-snippet) | Nonspecific | Neutral | Specific | Total |
| | 87 | 76 | 937 | 1,100 |

Furthermore, indicating the specificity classes of the antibodies mentioned in the snippets was made easier by color highlighting. The antibody snippets were populated to highlight the mentioned antibody in red and its specificity in blue, allowing for easier recognition of an antibody with its proposed specificity label.

Based on this specialized annotation interface with color highlighting, a curator would only need to continue reviewing each RRID snippet with its antibody snippet along the row to be able to adequately identify whether or not an RRID linking exists, as well as its identified specificity label. As such, the interface allowed for better and easier linking and specificity labeling practices moving forward. Two curators with antibody lab experience contributed to the linking and labeling of these snippets. Comparing the annotations by the two curators for 720 snippet pairs, we obtained $\kappa = 0.74$ for the task of linking and weighted $\kappa = 0.61$ for specificity. Weighted $\kappa$ was considered because the classes were ordered. Both are in the range of *substantial agreement*. These 720 pairs were selected from the error cases from our early versions of text mining systems and were relatively challenging to annotate. The annotations of all snippets were then cross-examined by other authors to make final annotations (Table 2).

The disagreement was mainly because initially "neutral" was not consistently annotated between curators, resulting in 128/720 cases being annotated as either "specific" or "neutral." Many of them state that a specificity test for antibodies was performed. If a positive result is reported then it is specific but neutral if no result is reported. For example:

*To confirm specificity of each monoclonal antibody, an ELISA was performed . . .*

The above sentence (PMID 27342872) is followed by a lengthy description of how the ELISA test was performed but no result was given and should be annotated as "neutral," though one may argue that no result given implies that no issue and thus is specific. However, we would like to train the model to classify based on given evidence instead of inferring what may be implied and decided to annotate in this way.

As expected, the data are highly skewed to the class "specific" because most authors validated their antibodies either experimentally or by citing one or more sources, but we still identified about 9.44% of cases that were nonspecific. It is expected that there should be more neutral statements but we filtered out most of them with keywords (missing any mention of terms like "antibody"). For the task of RRID linking, since specificity statements link only to a small number of all antibodies used in a study, "yes" linking constitutes only 15.18% of pairs. Finally, the last row of Table 2 shows the number of each specificity class of the specificity snippets annotated in the 1,100 positive RRID linking examples. These annotations served as our ground truth to evaluate the joint task—combining task 1 and task 2 to extracting triples of

RRID, specificity class, and the specificity snippet to populate the Antibody Watch knowledge base.

## Results

### Task 1: Specificity classification

Table 3 shows the results of 5-fold cross-validation for a broad range of models for Task 1 specificity classification. Our experiments comparing BERT, BioBERT and SciBERT showed that models using SciBERT constantly outperformed their counterparts though the results with the other two transformers are competitive. We therefore show the results of all models using SciBERT for an in-depth comparison and only the best models from BioBERT. The hyperparameter settings for the training are given in S1(C) Text.

Models named with "CLS" are those using Eq (2) as the final snippet representation, while those without use Eq (1). SciBERT and SciBERT-SPC are baseline models that use the output term of the last layer of the transformer as the predicted class without or with the aspect term (*i.e.*, "antibody"), respectively. We replaced the BERT layer in AEN [20] and LCF [19] with SciBERT and tested them with or without "CLS" as described above to create four competing models.

Among the twelve models tested, "AOA-CLS-SciBERT" in our $(ABSA)^2$ family achieved the best overall performance and the best for the nonspecificity (negative) class, the most important class for our aim of providing alerts of problematic antibodies. SciBERT-SPC also performed well as the overall second best model. Additional layers on top of the AOA model failed to improve the overall performance, though BiLSTM with CLS performed the best for the neutral class and can be helpful to exclude irrelevant statements. AEN and LCF models also performed well but fell short of our best $(ABSA)^2$ model. We note that our dataset is more unbalanced than the benchmark datasets for ABSA sentiment analysis [33] yet our numbers are at about the same level. The results suggest that our task of specificity classification can be accomplished with our $(ABSA)^2$ model.

**Table 3. Specificity classification performance comparison.**

| Model | Specific | Non-specific | Neutral | Macro F1 | Weighted F1 |
|---|---|---|---|---|---|
| $(ABSA)^2$ Models (Ours) | | | | | |
| AOA-SciBERT | 0.956 | 0.830 | 0.748 | 0.845 | 0.921 |
| **AOA-CLS-SciBERT** | **0.958** | **0.832** | 0.750 | **0.847** | **0.925** |
| AOA-BiLSTM-SciBERT | 0.954 | 0.820 | 0.734 | 0.836 | 0.918 |
| AOA-BiLSTM-CLS-SciBERT | 0.956 | 0.794 | **0.768** | 0.839 | 0.921 |
| AOA-MHA-SciBERT | 0.954 | 0.798 | 0.746 | 0.833 | 0.917 |
| AOA-MHA-CLS-SciBERT | 0.944 | 0.618 | 0.718 | 0.760 | 0.909 |
| SciBERT | 0.908 | 0.640 | 0.274 | 0.607 | 0.819 |
| SciBERT-SPC | 0.954 | 0.824 | 0.756 | 0.845 | 0.924 |
| BioBERT-SPC | 0.946 | 0.800 | 0.732 | 0.826 | 0.912 |
| AOA-BioBERT | 0.956 | 0.796 | 0.746 | 0.830 | 0.918 |
| AEN-SciBERT | **0.958** | 0.826 | 0.728 | 0.837 | 0.921 |
| AEN-CLS-SciBERT | 0.956 | 0.812 | 0.740 | 0.836 | 0.920 |
| LCF-SciBERT | 0.952 | 0.812 | 0.724 | 0.829 | 0.916 |
| LCF-CLS-SciBERT | 0.956 | 0.826 | 0.746 | 0.843 | 0.922 |

Numbers are F1; in bold fonts are the best results. Macro F1 is the average of F1 of all classes. **Weighted F1** is the average of F1 weighted by the number of instances of each class.

**Table 4. RRID-linking performance comparison.**

| Model | Precision | Recall | F1 | Accuracy |
|---|---|---|---|---|
| BERT-SPC | 0.830 | 0.859 | 0.844 | 0.952 |
| **SciBERT-SPC** | **0.861** | **0.896** | **0.878** | **0.962** |
| BioBERT-SPC | 0.844 | 0.874 | 0.858 | 0.956 |
| Baseline (0.8) | 0.579 | 0.633 | 0.605 | 0.483 |
| Baseline (0.9) | 0.591 | 0.665 | 0.626 | 0.600 |
| Baseline (1.0) | 0.603 | 0.671 | 0.635 | 0.689 |
| BiLSTM+Manhattan | 0.502 | 0.506 | 0.504 | 0.696 |
| BiLSTM+Euclidean | 0.522 | 0.536 | 0.529 | 0.664 |

Numbers in bold fonts are the best results.

## Task 2: Antibody RRID linking

A 5-fold cross-validation was applied to evaluate the performance of each model for the RRID linking task (Table 4), in which SciBERT-SPC performed the best, achieving 0.962 in accuracy, even though the data is highly skewed to "no" linking. Remarkably, the baselines and the BiLSTM Siamese models are far off from the level of the performance of those with transformers, suggesting that the task is difficult for these methods.

## Complete workflow

We have evaluated the performance for each individual task. We would like to evaluate both tasks jointly as a complete workflow shown in Fig 1. That is, how well can the workflow correctly extract triples of RRID, specificity class and snippet as the evidence to effectively populate the Antibody Watch knowledge base automatically?

Table 5 shows the result by joining our best performing models for Task 1 and 2 according to Tables 3 and 4, respectively, to assign RRID and specificity class to a specificity snippet. Here, a *true positive* is defined as a pair of RRID-snippet and specificity snippet where the RRID-linking assignment is "yes" and the specificity class is the same by both ground truth annotation and model prediction. *Precision* of a class $C$ is the ratio of the number of true positives of $C$ and the number of all predictions where the RRID-linking assignments are "yes" and the specificity classes are $C$. *Recall* of a class $C$ is the ratio of the number of true positives of $C$ and the number of all ground truth annotations where the RRID-linking assignments are "yes" and the specificity classes are $C$.

**Table 5. Complete workflow performance.**

| Class | Truth | Predicted | Precision | Recall | F1 |
|---|---|---|---|---|---|
| Nonspecific | 87 | 101 | 0.802 | 0.931 | 0.862 |
| Neutral | 76 | 81 | 0.728 | 0.776 | 0.752 |
| Specific | 937 | 924 | 0.938 | 0.925 | 0.932 |
| Total/Macro | 1,100 | 1,106 | 0.823 | 0.878 | **0.848** |
| Weighted | | | 0.913 | 0.915 | **0.914** |

The last second row shows the totals for the ground truth (Truth) and the predicted (Predicted) numbers and the macro averages of the metrics. The last row shows the weighted metrics as defined in the footnote of Table 3.

Table 5 shows that our workflow equipped with (ABSA)$^2$ for specificity classification and SciBERT-SPC for RRID linking achieved macro F1 over 0.8 and weighted F1 over 0.9 on 5-fold cross validation.

## Discussion

### Remarkable cases

When authors were unsure about the specificity of antibodies, we annotated those cases as nonspecific, *i.e.*, problematic. For example,

> *. . .A panShank antibody was also used to assay overall Shank protein levels, with the caveat that the affinity of the PanShank antibody to different Shank isoforms was unknown. . . .*

> (PMID 2925091)

Note that the snippet contains three sentences but we only show the sentence that illustrates the point to fit the page limit. Again, "PMID" is the PubMed ID of the paper where the example snippet appears.

A source of confusion that puzzled both curators and our deep neural network models is that the term "nonspecific" may be used to refer to an antibody that is used purposefully not to target an antigen in order to block unintended reactions or serve as a negative control in experimental validation of antibodies. For example, here is a snippet that should be labeled "positive."

> *. . .The supernatant was pre-cleared by immunoprecipitation with non $-$ specific antibodies (NIgG) to remove and identify non $-$ specific proteins, which may contaminate the Atk2 Co-IP and . . .*

> (PMID 26465754)

Another positive example with "nonspecific" mentioned:

> To test if this effect was direct, ChIP assays were performed using anti-E2F1 antibodies and specific primers amplifying the -214/+61 PITX1 proximal promoter region. . . .*A nonspecific antibody was used as a negative control, and the thymidine kinase (TK) promoter region, containing known E2F1 binding sites, was used as a positive control . . .*

> (PMID 27802335)

Still, most snippets with "nonspecific" state that an antibody does not always bind to its target antigen and is problematic.

> *. . .five out of six commonly used anti-Panx1 antibodies tested on KO mouse tissue in Western blots were "non$-$specific" . . .*

> (PMID 23390418)

(ABSA)$^2$ can correctly classify all examples above as well as double negation snippets as positive as given below.

> . . .*Negative controls for CD99 immunohistochemistry were established on adjacent lung sections by omitting incubation with CD99 primary* antibody *and did not demonstrate* nonspecific *binding.* . . .

(PMID 26709919)

## RRID

Our approach leverages RRIDs to link a snippet about antibody specificity to an exact antibody entity used in the reported study. Not all papers identify antibodies with their RRIDs. It is our plan to mine these statements with our task 1 model and manually (or semi-automatically) assign these statements to antibodies. We are currently developing a prototype pipeline to facilitate the whole process as a part of our dkNET service.

Meanwhile, the use of RRIDs has grown steadily. From an initial pilot in 2014 comprising approximately 25 journals, mostly in neuroscience, the use of RRIDs has grown considerably, with RRIDs appearing in over 1,200 journals and 21K articles. They are required by several major journals across multiple biomedical disciplines. Over 225K resources have been identified using the RRID specification as of March 2020. The RRID syntax was recently added to the Journal Article Tag suite (JATS ver# 1.3d1), an XML standard for mark-up of scientific articles in the biomedical domain, signaling that the academic publishing community has accepted RRIDs as a standard method for tagging research resources.

The prevalence of RRID and advances in NLP will allow text mining knowledge bases like Antibody Watch to grow into mature and indispensable references for scientists and general public users to obtain reliable statistics about a broad range of biomedical research resources. The approach presented here is an example where we add "landmarks" (*i.e.*, RRIDs) to allow automation to become practical. This approach can be traced back from the rails and airports for trains and airplanes to operate, to landmarks in an automobile plant for assembly robots to calibrate, and more recently, proposals to assign special traffic zones with signs and rules designed for autonomous vehicles. Mons [37] asked "why bury it first then mine it again?" and advocated the use of semantic tagging in scientific publication to facilitate biomedical text mining. Here, we present a successful use case where RRIDs serve as the landmarks for text mining robots, making detecting antibody specificity feasible and reliable.

## Future work

We plan to extend and replicate the general approach of integrating RRID with advanced deep neural network NLP models to quantify impact and influence of research resources by automated text mining. We will develop more text mining systems to extract quality statements about other research resources, including cell lines, data repositories, statistics and bioinformatics tools, *etc*. These statements will be made available through the "Resource Reports" developed by the NIDDK Information Network (dkNET.org). Resource Reports aggregate information based on RRIDs into a single report. Information includes basic information about the resource, papers that use the resource and any known issues regarding its performance. In a previous study [17], we showed that the use of RRIDs for cell lines correlated with a dramatic decrease in the number of papers published using problematic cell lines. We speculate that because when authors search the resource database for cell lines to obtain an RRID, they are confronted with a warning when the cell line is contaminated or misidentified. We hope to achieve the same type of alerting service for other types of resources like antibodies.

## Conclusion

In this work we present a novel approach to the antibody specificity problem by automated text mining and show that the approach is feasible. We formulated the problem and divided it into two tasks: 1) extracting and classifying snippets about antibody specificity and 2) linking the extracted snippets to antibodies that they refer to. We created a set of ground truth from two thousand articles reporting studies using antibodies as experimental reagents. We proposed an approach leveraging RRID to solve the challenging antibody identification problem and developed deep neural network models based on a transformer to achieve weighted F1 and accuracy over 0.9 for the two tasks respectively and the joint task when the two tasks combined to complete the workflow, where antibody specificity statements are extracted and assigned to exact antibody entities unambiguously. We will continue annotating more training examples to boost the performance. The approach can be applied to the ever growing number of publications and antibodies and provide scientists a reliable source about the specificity of antibodies.

## Supporting information

**S1 Text. Implementation details.** A. Overview of AEN and LCF. B. Siamese BiLSTM. C. Hyperparameter settings. Recently, models based on the attention-encoder network (AEN) and local context focus mechanism (LCF) models were reported to achieve state-of-the-art results for benchmark ABSA datasets. We also implemented these models and their variants for a comparison. AEN learns to build attention links within an input sentence and between the input sentence and the aspects to perform aspect-based sentiment analysis. LCF focuses on using attention mechanisms to relate local context to the aspects. The details of our implementation are given in A. A neural network model with a Siamese Recurrent Architecture consists of two recurrent neural network components trained to have identical parameters (thus the name "Siamese") to read a pair of different input sentences. Then the difference between the output vectors of the two components is computed to determine if the input sentences are similar. We used either Manhattan or Euclidean distance as the last layer to measure the similarity between the input snippet pair for our task of RRID linking. B describes the details. The hyperparameter settings for the training are given in C.
(PDF)

## Author Contributions

**Conceptualization:** Chun-Nan Hsu, Anita Bandrowski, Maryann E. Martone.

**Data curation:** Chun-Nan Hsu, Chia-Hui Chang, Thamolwan Poopradubsil, Amanda Lo, Karen A. William, Anita Bandrowski, Maryann E. Martone.

**Formal analysis:** Chun-Nan Hsu, Chia-Hui Chang, Thamolwan Poopradubsil, Maryann E. Martone.

**Funding acquisition:** Chia-Hui Chang, Anita Bandrowski, Jeffrey S. Grethe, Maryann E. Martone.

**Investigation:** Chun-Nan Hsu, Chia-Hui Chang, Thamolwan Poopradubsil, Ko-Wei Lin, Anita Bandrowski, Ibrahim Burak Ozyurt, Maryann E. Martone.

**Methodology:** Chun-Nan Hsu, Chia-Hui Chang, Thamolwan Poopradubsil, Amanda Lo, Karen A. William, Ko-Wei Lin, Anita Bandrowski, Ibrahim Burak Ozyurt, Maryann E. Martone.

**Project administration:** Ko-Wei Lin, Anita Bandrowski, Jeffrey S. Grethe, Maryann E. Martone.

**Resources:** Jeffrey S. Grethe, Maryann E. Martone.

**Software:** Chun-Nan Hsu, Chia-Hui Chang, Thamolwan Poopradubsil, Ibrahim Burak Ozyurt, Jeffrey S. Grethe.

**Supervision:** Chun-Nan Hsu, Chia-Hui Chang, Jeffrey S. Grethe, Maryann E. Martone.

**Validation:** Chun-Nan Hsu, Chia-Hui Chang, Amanda Lo, Karen A. William, Ko-Wei Lin, Anita Bandrowski, Maryann E. Martone.

**Visualization:** Chun-Nan Hsu, Chia-Hui Chang, Thamolwan Poopradubsil, Amanda Lo.

**Writing – original draft:** Chun-Nan Hsu, Chia-Hui Chang, Thamolwan Poopradubsil, Amanda Lo.

**Writing – review & editing:** Chun-Nan Hsu, Chia-Hui Chang, Thamolwan Poopradubsil, Ko-Wei Lin, Anita Bandrowski, Ibrahim Burak Ozyurt, Jeffrey S. Grethe, Maryann E. Martone.

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
