## [Decision Letter · Decision Letter 0]

10 Jan 2021

Dear Dr. Martone,

Thank you very much for submitting your manuscript "Antibody Watch: Text Mining Antibody Specificity from the Literature" for consideration at PLOS Computational Biology.

As with all papers reviewed by the journal, your manuscript was reviewed by members of the editorial board and by several independent reviewers. In light of the reviews (below this email), we would like to invite the resubmission of a significantly-revised version that takes into account the reviewers' comments.

We cannot make any decision about publication until we have seen the revised manuscript and your response to the reviewers' comments. Your revised manuscript is also likely to be sent to reviewers for further evaluation.

Sincerely,

Manja Marz

Software Editor

PLOS Computational Biology

Reviewer's Responses to Questions

**Comments to the Authors:**

Reviewer #1: Authors present a text mining-based tool to identify problematic antibodies by extracting statements regarding antibody specificity from the literature. The task is divided into two parts: Task 1 is to recognize antibody specificity snippets and labeling them as specific, non-specific, and neutral. The second task is to identify the RRID corresponding to the antibody from another text snippet, and combining them to generate antibody specificity triples. Neural network models are built for both tasks. An attention-on-attention model combined with SciBERT contextualized embeddings performs best for the first task, and a SciBERT-based sentence-pair model performs best for RRID linking task. The models are proposed for an antibody alerting knowledge base.

This manuscript reports a useful tool, and the knowledge base proposed would be a valuable resource for researchers working with antibodies. The manuscript is overall clearly written and well organized. The experimental work seems sound. Use of aspect-based sentiment analysis for the first task is sensible. Based on the examples given, the tasks (particularly the first) do not seem particularly complex (use of the words 'specific' and 'non-specific' seem quite predictive), but a couple of examples given in Discussion indicate that there can be some complex cases. I don't have a major problem with the manuscript, but I think it can improved in some aspects.

- It can be made clearer whether a snippet is always a sentence, or whether it can be multiple sentences or a sentence fragment.

- The data seems somewhat skewed due to their sentence selection criteria and may not be quite a representative sample. It would be interesting to see the results when applied to a randomly selected held-out dataset.

- The authors take the presence of RRID in a publication as a starting point. How much is being missed in terms of antibody specificity knowledge by making this choice? Could/should the approach be extended to cases/journals where RRIDs may not be used/enforced? Some discussion would be useful.

- Task 2 is cast as a sentence pair classification task. It is unclear how RRIDs are identified exactly once a pair is classified as positive. Does the RRID sentence/snippet in the pair always contain a single RRID? If multiple RRIDs are present in the snippet how is the selection done? Simply regular expressions? Any errors made here?

- Brief descriptions of some models (AEN, LCF, and Siamese Recurrent Architecture with BiLSTM) can be provided.

- BioBERT seems like a more natural choice for contextualized embeddings than SciBERT. I think it could be added to this paper, rather than mentioned as future work.

- What are the hyperparameters used to train the neural networks?

- A neural network architecture diagram could be useful.

- I was surprised that the inter-rater agreement was not better than "substantial", as the tasks seem relatively straightforward. Some examples of disagreement would be useful.

- It is not clear why task 2 involves 1100 sentences, when for Task 1 2639 sentences were labeled.

- Are antibodies always single tokens?

- RRID discussion on page 12 seems out of context and can probably be cut.

- Why does b = (h^L_n+2) start with n+2 instead of n+1?

- "classify such snippet into "- > snippets

- "models of baselines" -> baseline models?

- Give full name of BiLSTM when first introduced.

- concate -> concatenate

- "Note that a snippet contains three sentences": Is it "the snippet"?

- "is not always bind to" -> "does not always bind to"

- Figure 1 seems blurry, but should be fixed if accepted (Specifically the labels, e.g. PMC number, are unreadable).

Reviewer #2: 1. Because there is no benchmark dataset for this work, the authors developed a dataset for evaluation. However their dataset only include text containing the regex patterns of “(S|s)pecific, ((B|b)ackground staining)” or “(C|c)ross( |-)reactiv?”. Is it likely to lose some antibody specificity text, which cannot be identified by the patterns? The authors should clarify this point. What is the number of final papers in the dataset?

2. Each snippet contains both the previous and next sentences of the main sentence. Did the authors evaluate the effects of different numbers of surrounding sentences on performances?

3. How did the authors divide the dataset into 5-fold? Is it possible that a <rrid, antibody=""> pair appearing in the test fold also in the training fold?

4. It seems that their baselines for the RRID-linking can also apply to the Specificity classification. Is it possible to see the baselines on the specificity classification?

Minor:

1. Table 1, please add the PMIDs of examples.

2. Page 6, why do you mention "(RRID:SCR 018008)" at the 191st line? Also, Page 8, "RRID:SCR 017679" at the 252nd line.

3. Page 6, at the 181st line, the superscript "2" of "ABSA2" is part of your system name but looks like a footnote.

4. The text of Fig 1 is unclear.</rrid,>

**Have all data underlying the figures and results presented in the manuscript been provided?**

Reviewer #1: Yes

Reviewer #2: Yes

PLOS authors have the option to publish the peer review history of their article (what does this mean?). If published, this will include your full peer review and any attached files.

Reviewer #1: No

Reviewer #2: No
---

## [Decision Letter · Decision Letter 1]

15 Apr 2021

Dear Dr. Martone,

We are pleased to inform you that your manuscript 'Antibody Watch: Text Mining Antibody Specificity from the Literature' has been provisionally accepted for publication in PLOS Computational Biology.

Best regards,

Feilim Mac Gabhann, Ph.D.

Editor-in-Chief

PLOS Computational Biology

Jason Papin

Editor-in-Chief

PLOS Computational Biology

Reviewer's Responses to Questions

**Comments to the Authors:**

Reviewer #1: I thank the authors for their revisions. The manuscript is acceptable.

**Have the authors made all data and (if applicable) computational code underlying the findings in their manuscript fully available?**

Reviewer #1: None

PLOS authors have the option to publish the peer review history of their article (what does this mean?). If published, this will include your full peer review and any attached files.

Reviewer #1: **Yes: **Halil Kilicoglu

---

## [Editor Report · Acceptance letter]

21 May 2021

PCOMPBIOL-D-20-02034R1 

Antibody Watch: Text Mining Antibody Specificity from the Literature

Dear Dr Martone,

I am pleased to inform you that your manuscript has been formally accepted for publication in PLOS Computational Biology. Your manuscript is now with our production department and you will be notified of the publication date in due course.

With kind regards,

Zsofi Zombor
